# Prognostic Role of Cell Blood Count in Chronic Myeloid Neoplasm and Acute Myeloid Leukemia and Its Possible Implications in Hematopoietic Stem Cell Transplantation

**DOI:** 10.3390/diagnostics12102493

**Published:** 2022-10-14

**Authors:** Olga Mulas, Brunella Mola, Clelia Madeddu, Giovanni Caocci, Antonio Macciò, Giorgio La Nasa

**Affiliations:** 1Hematology Unit, Businco Hospital, ARNAS G. Brotzu, 09124 Cagliari, Italy; 2Department of Medical Sciences and Public Health, University of Cagliari, S554, km 4500, 09042 Monserrato, Italy; 3Department of Gynecologic Oncology, Businco Hospital, ARNAS G. Brotzu, 09124 Cagliari, Italy; 4Department of Surgical Sciences, University of Cagliari, 09124 Cagliari, Italy

**Keywords:** neutrophil-lymphocyte ratio (NLR), platelets, lymphocyte-monocyte ratio, monocyte, chronic myeloid leukemia, myeloid neoplasms, hematopoietic stem cell transplantation, prognosis

## Abstract

Numerous prognostic indexes have been developed in hematological diseases based on patient characteristics and genetic or molecular assessment. However, less attention was paid to more accessible parameters, such as neutrophils, lymphocytes, monocytes, and platelet counts. Although many studies have defined the role of neutrophil-to-lymphocyte or platelet-to-lymphocyte in lymphoid malignancies, few applications exist for myeloid neoplasm or hematopoietic stem cell transplantation procedures. In this review, we synthesized literature data on the prognostic value of count blood cells in myeloid malignancies and hematopoietic stem cell transplantation in the context of classical prognostic factors and clinical outcomes.

## 1. Introduction

A myeloid neoplasm (MN) refers to a heterogeneous group of hematological diseases affecting hematopoietic stem cells, including acute and chronic forms [1]. Different defects can contribute to the genesis of MN diseases, such as mutations in JAK2, CALR, MPL, FLT3 genes, or chromosome translocations. Therefore, different signaling pathways can be activated at the cellular level to promote an uncontrolled expansion of myeloid cells in the bone marrow [1,2]. The diagnosis of myeloid malignancy requires specific tests such as a trephine bone marrow biopsy and a morphology study of a myelogram [3]. Cytogenetic, molecular biology and cytofluorimetric analysis is also required [2]. The first exam required in hematology is a cellular blood count. (CBC). The test is an inexpensive and easy way to diagnose and monitor myeloid malignancies as well as other diseases.

The prognostic role of the distribution of the different cell populations in CBC has been assessed in different diseases, and in particular, the meaning of parameters such as neutrophil to lymphocyte ratio (NLR) and lymphocyte to monocyte ratio (LMR) has been explored [4,5,6]. High NLR has been shown to predict signals of severity in COVID-19 pneumonia [7] and mortality in coronary events [8]. Notably, NLR is an established parameter used for prognostic stratification in patients with solid tumors [9]. In the field of hematology, studies have evaluated the impact of these parameters focusing mainly on lymphomas [10]. However, the apparent alteration of CBC in myeloid diseases did not result in a debate about the proportion of different types of leukocytes. This review aims to assemble several studies that examined the role of NLR, LMR, platelets to lymphocytes ratio (PLR), absolute lymphocyte count (ALC), and absolute monocyte count (AMC) in myeloid malignancies. In addition, we discussed the role of those parameters in defining prognosis or response to therapy in most common myeloid diseases, such as chronic myeloid leukemia, Philadelphia-negative myeloproliferative neoplasm, myelodysplastic syndromes, and acute myeloid leukemia. Finally, we analyzed the effect of NLR, MLR, PLR AMC, and ACL on different aspects of hematopoietic stem cell transplantation (HSCT). Because of the great variability of the available data in HSCT, no distinction was made in terms of donor type, stem cell donor source or conditioning regimes.

## 2. Bone Marrow Microenvironment

The bone marrow microenvironment is located in the BM cavity, where hematopoietic cells can interact with multiple cellular components like osteoblasts, fibroblasts, mesenchymal cells, macrophages, and different matrix elements such as adhesion molecules, chemokines, cytokines, and soluble or membrane-bound factors. This relative barrier delimiting this microenvironment is also known as the BM niche [11]. 

The tumor microenvironment is a dynamical and complex milieu closely connected with all steps of carcinogenesis, composed of matrix and stromal cells, neuroendocrine, adipose, immune, and inflammatory cells, and lymphatic and vascular systems. Hematological malignancies present a slightly diverse microenvironment in comparison to solid tumors. The measurable parameters in the peripheral blood cells, such as neutrophils, lymphocytes, and platelets, may reflect the systemic inflammatory response associated with these tumor-induced microenvironment changes [4,6,9,12]. 

Indeed, the tumor microenvironment is complex and constantly evolving, and both adaptive and innate immune cells play a critical part in tumorigenesis [13]. Tumor progression is largely dependent on inflammation, which acts as a major driver with related oxidative stress, angiogenesis, matrix remodeling, and specific genetic mutations [14,15]. Similarly, in the hematologic field, alterations in the BM microenvironment have been demonstrated to represent a central step in the development of several myeloid malignancies and influence the peripheral CBC composition [16,17,18]. The composition of intra- and peri-tumoral immune cells affects both antitumor immunity and immunodeficiency. The percentage and location of different T cells influence the patient outcome: high intratumoral CD8+ cells are related to better outcomes [19,20], while a low amount of infiltrating CD8+ and CD4+ lymphocytes can promote relapse or metastasis [10]. Cells of the monocyte lineage are fundamental for the innate immune response and play a central role in the tumor microenvironment. In particular, tumor-associated macrophages (TAMs) can favor cancer cell proliferation, migration, and genetic instability and induce angiogenesis and lymphangiogenesis, which promote metastasis [21,22,23]. They can polarize into M1 or M2 phenotype, depending on different stimuli, which have specific pro- or anti-inflammatory profiles and exert different pro- or anti-tumoral actions [24]. A “reprogrammed” inflammatory BM microenvironment is a usual report in myeloid malignancies. Both malignant hematopoietic cells and proinflammatory cytokines are able to stimulate different categories of stromal cells in the bone marrow, induce their secretory activity, and affect their decreased hematopoiesis-supporting ability [10]. Macrophages and monocytes, as players of the innate immune response, are raised in chronic inflammatory conditions in myeloid malignancies. Other crucial cells are neutrophils, which exert anti-tumor activity but are also able to inhibit the cytotoxic action of lymphocytes. Moreover, granulocytes may foster cancer progression by stimulating changes in stromal cells and inducing the expression of specific cytokines, such as hematopoietic growth factor (HGF) and granulocyte colony-stimulating factor (G-CSF) [25,26].

Abnormally proliferating cancer cells in MPN can induce normal hematopoietic and stromal cells in the BM niche to release proinflammatory mediators, which can create chronic inflammation inside the niche [16].

Indeed, it has been described that the MPN BM stem cell niche is a place of chronic inflammation characterized by raised myeloproliferation, altered neutrophil apoptosis, and, probably, unbalanced neutrophil marginal and reserve pool. The specific alterations of these compartments are induced by the inflammatory stimuli and, in turn, can have a major impact on the resolution of inflammation in the MPN, both in the BM and in the peripheral blood, thus influencing the pathogenesis and progression of the disease [27]. Noteworthy, it has been reported that the JAK2 mutation, at the center of the MPN pathogenesis, activates the STAT3 signaling pathway, which is involved in a variety of inflammatory cytokines expressions causing inflammation and dysfunction of the immune system [28]. Hence the production of antibodies in anti-endothelial cells is associated with an increased risk of thrombosis [29]. In turn, inflammation may promote atheromatous damage and, therefore, the occurrence of thrombotic events, which are the major complication of MPN [30].

Then, high NLR can reveal a reduced number of lymphocytes and an increase of neutrophils in the tumor microenvironment. The absolute neutrophil count might act as an indicator of systemic inflammation, which favors tumor progression. Thus, also in myeloid diseases, NLR and MLR can express the connections between the tumor microenvironment and the host’s immune response and be strictly related to patients’ prognosis. 

## 3. Myeloid Malignancies

### 3.1. Chronic Myeloid Leukemia

Chronic myeloid leukemia (CML) is a clonal myeloproliferative expansion of transformed, primitive hematopoietic progenitor cells. Interestingly, BCR-ABL1, generated by a reciprocal translocation between chromosomes 9 and 22, has been linked to leukemia’s pathogenesis for the first time. In CML, myeloid progenitor cells expand at different stages of maturation, are released prematurely into the peripheral bloodstream, and settle extramedullary [31]. Since the 2000s, specific treatments such as tyrosine kinase inhibitors (TKI) have dramatically improved the disease’s outcome [32]. 

There is usually a strong correlation between white blood cell (WBC) count and disease prognosis in patients with CML. CML prognostic scores were first established in the CML based on, among other factors, the basophil and platelets count [33,34]. In contrast, low platelets are associated with higher mortality risk in the ELTS score [35,36]. The prognostic implications of ALC in CML were also investigated [37,38,39]. According to Sasaki et al., there were no significant differences in the cumulative incidence of complete cytogenetic response (CCyR) or molecular response (MR) at different time points. However, finding ALC ≥ 4000/µL at 3 or 6 months after TKI therapy was rare but associated with a decreased overall survival (OS) [37]. In patients treated with dasatinib, a significant increase in large granular lymphocytes (LGL) has been observed in peripheral blood. High levels of LGL were associated with a better response to therapy and excellent outcomes [38]. More recently, a report showed that ALC, AMC, and the LMR were not predictive of molecular response status in CML patients [39]. 

### 3.2. Myeloproliferative Neoplasm

Philadelphia-negative myeloproliferative neoplasms (MPNs) include three classical forms of clonal hematological malignancy: polycythemia vera (PV), essential thrombocythemia (ET), and primary myelofibrosis (PMF) [40]. Driver mutations causing MPN include JAK2, CALR, or MPL genes [41]. Clinical features are similar. Myelofibrosis can present either splenomegaly, cytopenias, or overlapping manifestations that lead to a more aggressive disease. Erythrocytosis occurs most often in PV, while thrombocytosis occurs in ET [42]. 

It is also possible for patients with ET and PV to progress toward myelofibrosis. A thrombotic event and disease progressions are the most severe complications and the leading causes of death in patients with MPNs [43,44,45]. 

Prognostic scores are based on age and previous thrombotic events [46,47]. The international prognostic score for ET (IPSET) included leukocyte ≥ 11 × 10^9^/L as a parameter associated with a higher risk of thrombotic events [47]. Studies have investigated the role of NLR in preventing ET thrombosis [48,49,50]. According to Hacibekiroglu et al., erythrocyte sedimentation rate, C-reactive protein, RDW, MPV, and NLR in the genesis of thrombosis, are commonly used chronic inflammation indicators [48]. There were significant differences in the main laboratory results between healthy controls and patients. No significant differences were found in the patient cohort between those with and without thrombotic events [48]. 

An analysis of 70 PV and ET patients found no association among thrombosis, PLR, and NLR [49]. However, in a larger group of 150 ET patients, NLR was the best predictor of thrombosis events. It also reported a high NLR ratio in Jak2-positive patients [50]. Additionally, high NLR was found in patients at high-risk stratification of thrombosis [50]. In PV, leukocytosis played a critical role over time. Indeed, several authors have reported a close relationship between leukocytosis at diagnosis and poor prognosis or high risk of thrombosis [51,52,53]. According to Boiocchi et al., patients with persistent leukocytosis have worse outcomes than those without [54]. In a similar study, Ronner et al. found that persistently elevated leukocytes were associated with increased disease evolution but not thrombotic events [55]. 

The above suggests that biomarkers like NRL and PLR calculated at disease diagnosis might be a better and easier way of predicting disease outcomes. A recent editorial demonstrated that NRL higher than 3.48 had a shorter time to thrombosis (TTT), and NLR > 2.62 was associated with the lowest OS. Similarly, patients with a high (>148.8) PLR had an inferior OS, and a higher level of PLR > 210.68 was associated with a lower TTT [56]. In addition, the increased value of NLR was an independent predictor of venous thrombosis in a recent revision of the ECLAP trial [57]. Venous thrombosis events were recently associated with a high absolute neutrophil count (ANC) and AMC at diagnosis of MPN [58]. In particular, V617F% ≥ 75% or AMC ≥ 1.5 × 10^9^/L was found to be strongly associated with a higher risk for venous thrombosis in Post-PV MF [59].

MF shows the most aggressive biological behavior among the Ph-MPNs and is associated with the highest mortality rate [60]. Historically, hemogram parameters such as WBC, hemoglobin, or platelets have historically been used to calculate MF scores [61,62,63]. Nevertheless, few studies have examined the correlation between NLR or PLR and MF outcomes [64,65]. There was a significant association between higher NLR and Jak2 mutation, such as ET [64]. A greater NLR and PLR were found in MF patients than in the general population. Among MF patients with higher PLR, a less aggressive disease was revealed with the absence of blast phase disease, constitutional symptoms, smaller spleen size, and lower CRP [64]. Another study showed that low ALC in patients with MF could predict inferior survival [65]. 

### 3.3. Myelodysplastic Syndrome

The myelodysplastic syndrome (MDS) is a hematological clonal neoplasm in which ineffective hematopoiesis occurs in one or more blood cell lineages due to dysmorphogenesis. As a result, there were one or more cytopenias [66]. As the disease progressed, prognostic scores were developed on bone marrow blast percentage, karyotype, degree of cytopenia, red blood cell transfusion need, age, and performance status [67,68]. 

The higher-risk MDS patients are more likely to undergo HSCT because of the risk of leukemic transformation and the very short OS, even without transformation [66]. Adaptive immunity has been linked with an increased interest in MDS prognosis. A different distribution of CBC parameters was observed according to the type of MDS. Notably, high levels of ACL were associated with a specific subgroup of MDS patients with ring sideroblasts, known to have a good prognosis [69]. MDS of multilineage dysplasia, on the other hand, was associated with a lower level of PLR [70]. However, lymphocyte counts were low in therapy-related MDS. In patients with higher-risk MDS (IPSS-R intermediate, high, and very high), there were lower ALC levels compared to lower-risk MDS [69]. 

Among low-risk MDS patients, an ALC below 1.2 × 10^9^/L was an additional negative prognostic factor. In accordance with the above-mentioned studies, ALC levels < 1.2 × 10^9^/L were associated with poor OS [71,72,73]. Saeed et al. also found that low levels of monocytes were correlated with poor OS in MDS [72]. Following treatment, low baseline neutrophil, monocyte, and lymphocyte levels were associated with an increased risk of infection or bacteremia [74]. Finally, few data are available between CBC and response to treatment. In this context, a low platelet count after therapy may indicate poor response [75]. 

### 3.4. Acute Myeloid Leukemia

Acute myeloid leukemia (AML) is a hematological malignancy characterized by the proliferation of abnormal clonal hematopoietic precursors. AML usually occurs de novo in healthy individuals, but it can be triggered by a different hematological disease or from previous chemotherapy or radiotherapy exposure [76]. The clinical condition can manifest as cytopenias or hyperleukocytosis, and it can progress very rapidly. The appropriate therapy depends on the patient’s age, fitness, and prognostic risk score [77,78].

Furthermore, the prognostic importance of CBC has been demonstrated in several studies [79,80,81]. There is a possibility that platelet levels are prognostic in AML. Patients with medium platelet count (50–120 × 10^9^/L) had longer OS and disease-free survival (DFS) than those with a low or high platelet count (<50 × 10^9^/L or >120 × 10^9^/L, respectively) [79]. It appears, however, that treatment response is more variable. Non-response rates were higher in patients with low platelet counts at 21 days after induction treatment [75]. Similarly, a high platelet count at day 14 predicted better outcomes in elderly patients [80]. In contrast, Zhang Y. et al. recently found that for intermediate-risk AML patients treated with chemotherapy, a platelet count < 40 × 10^9^/L was associated with better OS and DFS [81]. 

Studies focusing on lymphocytes and monocyte cells showed that low AMC and ALC correlate with an increased risk of infectious complications but not death [74]. Monocyte count can predict response and OS in AML patients at different stages of the disease [82]. 

For instance, Ismail et al. found that an AMC ≥0.8 × 10^9^/L at day 28 after therapy was associated with short OS and DFS [83]. Moreover, a high level of NLR at diagnosis and relapse has been associated with a poor prognosis [84,85]. However, data on the role of ALC are contradictory. According to some authors, higher ALCs assessed before and after the first chemotherapy have poor predictive value [86,87,88,89]. A large study of 1702 AML patients showed that an ALC threshold lower than 1 × 10^9^/L was associated with poor OS and DFS [88]. Other authors have reported better OS and DFS in patients with ALC > 0.35 × 10^9^/L 28 days after starting treatment [83]. Similarly, Keenan et al. found that higher ALC correlates with better OS after each chemotherapy course, even though the difference is more pronounced after induction [90]. As confirmed by additional research, even the HSCT does not seem to improve the prognosis of patients with ALC < 500 × 10^9^/L that had lower OS after chemotherapy [91]. 

Table 1 and Table 2 summarize the main papers referred to in the article about myeloid malignancies. 

## 4. Hematopoietic Stem Cell Transplantation

Hematopoietic stem cell transplantation (HSCT) has been considered one of the most effective and sometimes the only curative options in the treatment of myeloid hematologic malignancies for decades [92]. HSCT is commonly recommended for CML patients who have failed two or more TKI lineage, but for Ph-negative myeloproliferative neoplasms, it is less indicated [43,93,94]. In contrast, high-risk MDS and AML are the majority of patients to receive this treatment [95,96,97]. 

The clinical decision to perform an HSCT in these patients is based on balancing the risk of transplant-related mortality, the availability of a suitable donor, and the mortality associated with disease progression and evolution [97,98]. HSCT consists of several phases. The conditioning regime is required to destroy residual hematopoietic activity disease. Following the infusion of the donor, the HSCs engraftment is expected [99,100]. HSCT can be complicated by infections, conditioning-related toxicity, graft failure, and acute or chronic graft versus host disease (GVHD) [98]. Moreover, it can help control minimal residual disease (MRD) as a result of the well-known graft versus leukemia effect (GVL) [101]. Some authors have recommended monitoring the prognosis of HSCT based on CBC. Despite AMC > 0.3–0.5 × 10^9^/L after HSCT being associated with better survival, a more consistent result was seen when lymphocyte subsets were examined [102,103]. Studies have shown that a minimum level of ALC of 300 × 10^9^/L or higher improves OS and is associated with fewer infectious complications after transplantation [104,105,106,107,108,109,110,111]. GVHD is a major cause of morbidity and mortality following allogeneic hematopoietic cell transplantation [112]. ALC and eosinophil count (EC) at the time of chronic GVHD were incorporated into the chronic graft-versus-host disease risk score (CIBMTR). ALC and EC below normal were proven to contribute to adverse OS outcomes [113]. 

The use of biological drugs to prevent GVHD can affect the outcome of this procedure. Those with high levels of ALC had a worse prognosis when starting the immunomodulating treatment with alemtuzumab [114]. Moreover, higher levels of ALC predict better OS in patients who received high doses of anti-thymocyte globulins (ATG) [115]. Similarly, in a recent report, patients with preconditioning ALC < 500 × 10^9^/L were associated with short OS and higher infectious mortality due to a side effect of excessive doses of ATG and profound T-cell depletion [116]. ALC > 150 × 10^9^/L before ATG infusion correlated with higher rates of acute GVHD requiring steroids and non-relapse mortality [117]. On the contrary, the slow lymphocyte recovery after HSCT (<200 × 10^9^/L) could suggest a change in GVHD treatment based on a higher risk of disease relapse [118,119,120]. Nevertheless, Afzal et al. did not find any correlation between early lymphocyte recovery and the graft versus leukemia effect in pediatric AML patients [121]. In addition, to immune cells, other specific types of immune cells were investigated in HSCT. Natural killers (NK) are among the first type of immune cells to recover after HSCT and are thought to contribute to the graft versus leukemia effect [122]. Indeed, the level of NK was found to be low in patients who relapsed from CML after HSCT [123]. In acute leukemia transplants, higher NK levels were significantly associated with better DFS and TRM than patients with NK below 120/µL [124]. More recently, Minculescu et al. reported that NK >150/mm^3^ on day +30 predicts better OS, lower TRM, and infections [125]. Table 3 shows the most important articles in this field.

## 5. Discussion

Complete blood counts are crucial to diagnosing, determining prognosis, and monitoring treatments in hematological malignancy [2,40,42,43]. Thus, distinguishing between changes related to the tumor burden and those related to inflammation is complex but crucial. Compared with the general population, myeloid malignancy exhibited a high NLR, PLR, and AMC [48]. In myeloproliferative neoplasm, high leukocyte levels and NLR were found in patients with thrombotic complications [56]. NLR was an independent predictor of venous thrombosis in PV patients [57]. Overall, this results in a lower OS [56]. Further, inflammation can promote thrombosis, and platelets and neutrophils can contribute together [126,127].

Moreover, an inflammatory BM microenvironment with specific changes in the cells of the monocyte-macrophage lineage and the associated release of different proinflammatory cytokines is able to influence not only the development and progression of MPN by inducing fibrosis, thrombosis, tumor angiogenesis, and metastasis, but also the proportion of pro-tumoral/anti-tumoral immune cells both locally and systemically [16,18]. Thus, peripheral CBC parameters may reflect the peculiar changes of BM microenvironment and the status of associated chronic inflammation, which is emerging as a major driver of MPN evolution, and thus represent significant prognostic parameters.

The variation in immune cell expression defines disease manifestation and treatment response. Lymphocytes are the most influential of these cells. Lymphocyte deficiency can allow cancer cells to escape immune system control [128]. In solid tumors, a low amount of lymphocyte infiltration is associated with relapse or metastasis [129,130]. In CML, the level of circulating LGL is correlated with better outcomes in patients receiving dasatinib treatment [38]. MDS and AML patients with lower levels of ALC have been associated with a poorer prognosis of the disease and shorter OS [71,72,73,91,131]. In the transplant setting, ALC prognostic role has been thoroughly investigated and correlated with the immune response reconstitution, a pivotal role after hematopoietic stem cell transplantation [106,107,110,132].

Several studies have concluded that lymphocyte immune reconstitution begins with NK cells in the first month after HSCT, followed by CD8+ and B, and finally by CD4+ cells, improving the patient’s defense against pathogens. The process can take up to two years [105,132]. Thus, ALC at + 21 and +30 can offer a good assessment of the ability of the donor NK cells to control the residual disease [102,104]. Their killing action is well defined, and a low level in CML transplanted patients is associated with relapse of disease [123].

Furthermore, better lymphocyte reconstitution correlates with better infection control, leading to higher DFS and OS rates. The ALC level affects the response to immunomodulating treatment in conditioning regimens and correlates with a better prognosis and a lower TRM [102,104,124]. There is also a delicate balance between immunological recovery and the risk of complications from GVHD in HSCT [132]. In fact, lymphocytes can contribute to GVHD complications, leading to significant morbidity and mortality rates [117]. Dysregulation of the cell pattern may predict the risk for GVHD, especially when a high CD4/CD8 ratio is found [124].

The investigation of the prognostic role of different CBC parameters, such as NLR, ALC, LMR, and PLR, in myeloproliferative neoplasm, represents an intriguing area of research. The definition of their actual clinical significance may help classify the prognosis of patients affected by these hematological malignancies more precisely. Even in HSCT, those parameters could be significant in predicting the risk of complications such as TRM and GVHD. However, there is currently not enough data to define its specific role. Furthermore, many variables, such as conditioning regimen, donor type, and CSE sources, may improperly influence their interpretation.

## 6. Conclusions

The predictive and protective role of NRL, PLR, or absolute count of lymphocytes and monocytes in myeloid malignancy has been poorly evaluated. Clearly, this is a simple tool to obtain and may have significant effects. Additional studies are needed.

## Figures and Tables

**Table 1 diagnostics-12-02493-t001:** Prognostic role of peripheral count blood cells parameters in chronic myeloproliferative neoplasms.

Authors	Years	Diseases	No. of Pts	Outcomes	Comments
Sasaki [37]	2014	CML	483	OS	ALC ≥ 4 × 10^9^/L at 3 or 6 months of TKI start was associated with lower OS
Pepedil-Tanrikulu [39]	2020	CML	95	Response	ALC, AMC, and LMR did not predict the molecular response
Hacibekiroglu [48]	2015	ET	99	Thrombosis	No differences in CRP, NLR, RDW, MPV, and sedimentation levels
Kocak [49]	2017	ET	70	Thrombosis	No differences in NLR and PLR in patients with or without thrombosis
Zhou [50]	2018	ET	150	Thrombosis	Higher NLR in Jak2 positive and in patients at high-risk stratification of thrombosis
Boiocchi [54]	2015	PV	10	Evolution	Persistent hyperleukocytosis is associated with poor prognosis in MF-post-PV pts
Ronner [55]	2020	PV	520	Evolution, thrombosis	Persistently elevated leukocyte was associated with an increased hazard of disease evolution but not of thrombotic events
Krečak [56]	2021	PV	109	OS, thrombosis	Higher NRL and PLR are associated with a high risk of disease, shorter TTT
Carobbio [57]	2021	PV	1508	Thrombosis	High NLR is an independent predictor of venous thrombosis
Farruk [58]	2022	PV, TE	487	Thrombosis	ANC and AMC associated with venous thrombosis
Teng [59]	2022	Post-PV MF	163	Thrombosis	Patients with V617F% ≥ 75% or AMC ≥ 1.5 × 10^9^/L had a higher risk for venous thrombosis
Lucijanic [64]	2018	MF	102	OS	Higher NLR and Jak2 mutation; High NLR and low PLR poor prognosis
Lucijanic [65]	2018	MF	83	OS	Low ALC associated with poor prognosis

Abbreviations: OS, overall survival; TTT, time to thrombosis; CML, chronic myeloid leukemia; ET, essential thrombocythemia; PV, polycythemia vera; MF, myelofibrosis; ALC, absolute lymphocyte count; ANC, absolute neutrophil count; TKI, tyrosine kinase inhibitors; AMC, absolute monocyte count; NLR, neutrophil-to-lymphocyte ratio; LMR, lymphocyte-to-monocyte ratio; CRP, C-reactive protein; RDW, Red blood cell distribution width; MPV, mean platelet volume; PLR, platelets-to-lymphocytes ratio.

**Table 2 diagnostics-12-02493-t002:** Prognostic role of peripheral count blood cell parameters in acute myeloid leukemia and myelodysplastic syndromes.

Authors	Years	Diseases	No. of Pts	Outcomes	Comments
Silzle [58]	2019	MDS	1023	OS	Low ALC and lower OS, most apparent in lower-risk patients
Yikilmaz [70]	2020	MDS	63	Classification	Low PLR and multilinear dysplasia
Jacobs [71]	2010	MDS	503	OS	ALC > 1.2 × 10^9^/L better OS
Saeed [72]	2017	MDS	889	OS	Low ALC, low AMC, inferior OS. ALC, AMC, and LMR are not influenced by LFS
Saeed [73]	2016	MDS	889	OS, LFS	Low ALC lower OS but not lower LFS, most apparent in lower-risk patients
Buckley [74]	2014	MDS/AML	205	Complication	Low AMC and ALC at induction treatment high risk of infection or bacteremia
Chen [75]	2015	AML/MDS	343	Response	Low PLT counts at 21 after induction was associated with no response
Zhang [79]	2017	AML	209	OS, DFS	PLT between 50 and 120 × 10^9^/L better OS and DFS
Huang [80]	2018	AML	117	Prognosis	PLT count recovery on day 14 after D-CAG IC is associated with the response
Zhang [81]	2020	AML	291	OS and DFS	Low platelets levels at diagnosis predict better OS and DFS
Feng [82]	2016	AML	193	OS	High AMC appeared as a poor prognostic factor for OS
Ismail [83]	2019	AML	83	OS, DFS	AMC ≥ 0.8 × 10^9^/L + 28 shorter OS and LFS, ALC > 0.35 × 10^9^/L higher OS and LFS
Zhang [84]	2021	AML	181	OS, DFS	NLR < 2 at diagnosis better OS and DFS
Mushtaq [85]	2018	AML	63	OS	High NLR independently predicts poor OS in RR-AML patients.
Lobanova [86]	2017	AML	35	DFS	ALC more than 0.8 × 10^9^/L poor DFS
Jang [87]	2019	AML	65	LFS, OS	Higher ALC poor LFS and OS
Le Jeune [88]	2013	AML	1702	OS, DFS	Initial ALC < 1 × 10^9^/L poor DFS and OS, ALC > 4.5 × 10^9^/L lower response rate IC
Bar [89]	2015	AML	259	OS	Higher ALC lower remission and poor RFS and OS
Keenan [90]	2012	AML	59	OS	At +28 days post IC ALC > 1.35 × 10^9^/L better OS
Bumma [91]	2014	AML	180	OS	ALC < 0.5 × 10^9^/L poor outcome in IC

Abbreviations: MDS, myelodysplastic syndrome; AML, acute myeloid leukemia; OS, overall survival; LFS, leukemia-free survival; DFS, disease-free survival; IC, induction chemotherapy; D-CAG, decitabine, cytarabine, aclarubicin, and granulocyte colony-stimulating factor; ALC, absolute lymphocyte count; AMC, absolute monocyte count; PLR, platelet-to-lymphocyte ratio; NLR, neutrophil-to-lymphocyte ratio; PLT, platelets; RFS, relapse-free survival.

**Table 3 diagnostics-12-02493-t003:** Prognostic role of different peripheral count blood cell parameters in bone marrow transplantation.

Authors	Years	No. of Pts	Outcomes	Comments
Thoma [102]	2012	135	OS	ALC and AMC > 0.3 × 10^9^/L from +30 +60 and +100 from HSCT had better OS
Tang [103]	2018	59	OS	AMC > 0.57 × 10^9^/L +15 from HSCT had better OS
Chang [104]	2013	78	OS, TRM	ALC > 0.3 × 10^9^/L lower relapse rates and lower infections. Better OS, LFS, and low TRM
Bayraktar [105]	2015	518	OS, NRM	ALC > 0.3 × 10^9^/L at +60 better OS and NRM after HSCT
Chakrbarti [106]	2003	29	OS, NRM	ALC > 0.3 × 10^9^/L at +30 was the strongest predictor of NRM and OS
Fu [107]	2016	134	OS, LFS	ALC > 0.294 × 10^9^/L at +30 better OS and LFS, but was not related to relapse
Gul [108]	2015	381	OS, NRM	ALC < 0.4 × 10^9^/L lower OS and increased NRM. No association with relapse
Han [109]	2013	69	OS, EFS	ALC > 0.5 × 10^9^/L at +21 and +30 better engraftment. High ALC at 30 days had better OS and EFS. There were no differences in the GVHD or relapse rates.
Porrata [110]	2002	45	OS	ALC > 0.5 × 10^9^/L at day +15 had better OS
Le Bourgeois [111]	2016	47	OS	+30 ALC > 2.76 × 10^9^/L and +42 ALC > 4.25 × 10^9^/L had better OS.
Moon [113]	2017	307	cGVHD	ALC < 1 × 10^9^/L and eosinophil count < 0.5 × 10^9^/L relate to lower OS and helped improve the risk stratification power of CIBMTR
Sheth [114]	2019	364	OS, DFS	ALC > 0.08 × 10^9^/L in 2 days of alemtuzumab infusion had poor DFS and OS
Kennedy [115]	2018	135	OS	High ALC in higher recipient ATG dose had a lower risk of death
Seo [116]	2021	64	OS	PC ALC < 0.5 × 10^9^/L shorter OS and higher infectious mortality in patients receiving ATG
Shiratori [117]	2021	53	GVHD	ALC > 0.15 × 10^9^/L before ATG predicts GVHD requiring systemic steroids
Kumar [118]	2001	87	OS, relapse	ALC < 0.15 × 10^9^/L at +30 lower OS and higher relapse rates.
Powles [119]	1998	201	Relapse	ALC > 0.2 × 10^9^/L associated with lower relapse rates
Michelis [120]	2014	191	OS, relapse	ALC > 0.5 × 10^9^/L +28 is associated with lower relapse
Afzal [121]	2009	71	EFS relapse	ALC does not correlate to GVL and was not predictive of relapse in AML children
Jiang [123]	1997	15	GVHD	No correlation between CD4c, CD8c, or NK cells and the development of GVHD
Huttunen [124]	2015	83	GVHD, EFS	CD4/CD8 higher in patients with GVHD, NK > 0.12 × 10^9^/L at +32 had better TRM and EFS
Minculescu [125]	2016	298	OS, relapse	NK > 0.15 × 10^9^/L on +30 had better OS less TRM and infections. No link to relapse.

Abbreviations: OS, overall survival; EFS, event-free survival; DFS, disease-free survival; NRM, non-relapse mortality; TRM, transplant-related mortality; GVHD, graft versus host disease; ALC, absolute lymphocyte count; AMC, absolute monocyte count; CIBMTR, chronic graft-versus-host disease risk score; GVL, graft versus leukemia; NK, natural killer; HSCT, hematopoietic stem cell transplantation.

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
