# Peer review of "Prognostic Role of Cell Blood Count in Chronic Myeloid Neoplasm and Acute Myeloid Leukemia and Its Possible Implications in Hematopoietic Stem Cell Transplantation"

_diagnostics, 2022, doi:10.3390/diagnostics12102493_

Round 1
Reviewer 1 Report
Feedback for Mulas et al., 2022
The review article by Mulas et al, titled “ Prognostic role of cell blood count in myeloid neoplasms” summarizes the prognostic value of blood cell count in myeloid neoplasms (CML, Ph- MPNs, MF, MDS and AML) and the outcome of hematopoietic stem cell transplantation (HSCT). This article allows readers to appreciate parameters including neutrophil-to-lymphocyte ratio (NLR), lymphocyte to monocyte ratio (LMR) and platelet-to-lymphocyte ration (PLR) in as important prognostic tools for myeloid malignancies and HSCT.
The major and minor critiques are listed below.
Major points:
1. The scope of the review might be too broad. Myeloid neoplasms encompass a list of disease indication that can be found in “The World Health Organization (WHO) classification of the myeloid neoplasms” by Vardiman et al in 2002. My suggestion is to focus on 1 or 2 particular indications; either: a. primary AML and therapy related AML or b. MDS and therapy related MDS.
2. Please justify the review of the prognostic role of different peripheral count blood cells parameters in bone marrow transplantation (BMT) and not prognostic role of different peripheral count blood cells parameters in of PBSC transplantation and cord blood transplantation.
For example, in addition to BMT, MDS patients do receive PBSC transplant from unrelated or unrelated donors.
3. Differences NLR, LMR and PLR is not only attributed to the type of disease, but also the conditioning regiment and the type of immunosuppressants used as GvHD prophylaxis. Please explain the insufficient coverage of this aspect in the review article
Minor point:
Due to the broad scope of the review a total of 121 articles were reference. In many instances summaries were based on 1 reference article. For example, in page 5 of 20 (line 208 to 209) “Patients with medium platelet count has longer OS and disease-free survival )DFS) than those with a low of high platelet count (ref 69). Please define the parameters for low/medium/high platelet count.

Author Response
Point-by-point reply to Reviewers’ comments
Reviewer 1 Comments and Suggestions for Authors
Feedback for Mulas et al., 2022
The review article by Mulas et al, titled “ Prognostic role of cell blood count in myeloid neoplasms” summarizes the prognostic value of blood cell count in myeloid neoplasms (CML, Ph- MPNs, MF, MDS and AML) and the outcome of hematopoietic stem cell transplantation (HSCT). This article allows readers to appreciate parameters including neutrophil-to-lymphocyte ratio (NLR), lymphocyte to monocyte ratio (LMR) and platelet-to-lymphocyte ration (PLR) in as important prognostic tools for myeloid malignancies and HSCT.
The major and minor critiques are listed below.
Major points:
- The scope of the review might be too broad. Myeloid neoplasms encompass a list of disease indication that can be found in “The World Health Organization (WHO) classification of the myeloid neoplasms” by Vardiman et al in 2002. My suggestion is to focus on 1 or 2 particular indications; either: a. primary AML and therapy related AML or b. MDS and therapy related MDS.
Reply: Dear Reviewer, we appreciate your comment. The purpose of this review was to include in a single paper all the articles that analyzed the role of CBC parameters in chronic and acute myeloid malignancy and bone marrow transplant procedures. Indeed, we found that this topic is discussed in hematology less frequently than in solid neoplasms, and, for this reason, we believe it might be helpful to find the main articles published on the topic in the same paper including the main disease according to the myeloid neoplasm classification (ref 2). Moreover, considering your suggestion to better clarify yet in the title the disease covered by the paper, we propose to change the title from “Prognostic role of cell blood count in myeloid neoplasms” to “Prognostic role of cell blood count in chronic myeloid neoplasm and acute leukemia and its possible implications in hematopoietic stem cell transplantation”
- Please justify the review of the prognostic role of different peripheral count blood cells parameters in bone marrow transplantation (BMT) and not prognostic role of different peripheral count blood cells parameters in of PBSC transplantation and cord blood transplantation.
For example, in addition to BMT, MDS patients do receive PBSC transplant from unrelated or unrelated donors.
Reply: Given the limited number of papers available, we considered the HSCT generically including all forms, but without considering differences between the various CSE sources, types of conditioning regimes, or donor types. Please find added on line 53 the following sentence “Because of the great variability of the available data in HSCT, no distinction was made in terms of donor type, stem cell donor source or conditioning regimes.” To avoid confusion, I have also changed the title of the corresponding paragraph into “Hematopoietic stem cell transplantation in accordance with the actual content reported in the text.
However, there are currently too many variables which may affect these parameters. Differences NLR, LMR and PLR is not only attributed to the type of disease, but also the conditioning regiment and the type of immunosuppressants used as GvHD prophylaxis. Please explain the insufficient coverage of this aspect in the review article
Reply: We cannot draw clear conclusions about the predictive role of PLR or LMR in the HSCT. The main reasons for this are the few studies available, and the heterogeneity with which CBC-related data are reported. However, summarizing the literature using the schematic table can allow the reader to critically evaluate the data reported. Please find the following sentence on lines 395-399. “Even in HSCT, those parameters could be significant to predict the risk of complications such as TRM and GVHD. However, currently, there is not enough data available to define its specific role. Furthermore, many variables may improperly influence their interpretation such as conditioning regimen, donor type and CSE sources”
Minor point:
Due to the broad scope of the review a total of 121 articles were reference. In many instances summaries were based on 1 reference article. For example, in page 5 of 20 (line 208 to 209) “Patients with medium platelet count has longer OS and disease-free survival )DFS) than those with a low of high platelet count (ref 69). Please define the parameters for low/medium/high platelet count.
Reply: thank you for your suggestion. I have checked and added the specific references in different sentences (the final reference count in the revised version is 132). In details, as regard the specific example cited by you at page 5, we have changed the sentence from “Patients with medium platelet count had longer OS and disease-free survival (DFS) than those with a low or high platelet count” to “Patients with medium platelet count (50–120 × 109 /L) had longer OS and disease-free survival (DFS) than those with a low or high platelet count (<50× 109/L or >120 × 109/L respectively)” (see lines 222-225 of the revised version)
Reviewer 2 Report
I read with interest and curiosity the review entitled “Prognostic role cell blood count in myeloid neoplasms” as it deals a topic not very present in the literature. I think that this article has originality and it is well-written. The findings and conclusions are appropriate and in according to the project research. I have only one suggestion about the thrombosis in the MPNs. Infact, it has been reported that the JAK2 mutation activates the STAT-3 pathway signaling causing inflammation and dysfunction of the immune system and hence the production of antibodies anti-endothelial cells associated with increased risk of thrombosis. Are there data in literature about the cell blood count such as ACL or AMC and thrombosis in MPNs? This information can give an further boost of novelty to this review and provide a new clinical marker of thrombotic risk in the MPNs. I think that this information should be reported in this manuscript. Therefore, I think that this article is suitable for publication after minor revision.
Author Response
Point-by-point reply to Reviewers’ comments
Reviewer 2
Comments and Suggestions for Authors
I read with interest and curiosity the review entitled “Prognostic role cell blood count in myeloid neoplasms” as it deals a topic not very present in the literature. I think that this article has originality, and it is well-written. The findings and conclusions are appropriate and in according to the project research.
I have only one suggestion about the thrombosis in the MPNs. In fact, it has been reported that the JAK2 mutation activates the STAT-3 pathway signaling causing inflammation and dysfunction of the immune system and hence the production of antibodies anti-endothelial cells associated with increased risk of thrombosis.
Reply: Thank you very much for your comment. I really appreciate your comments. Please find the additional sentence in lines 105-111 “Noteworthy, it has been reported that the JAK2 mutation, which is at the center of MPN pathogenesis, activates the STAT3 signaling pathway, that is involved in a variety of inflammatory cytokines expression causing inflammation and dysfunction of the immune system [28] and hence the production of antibodies anti-endothelial cells associated with increased risk of thrombosis [29]. In turn, inflammation, may promote atheromatous damage, and therefore the occurrence of thrombotic events, which are the major complication of MPN [30].”.
Are there data in literature about the cell blood count such as ACL or AMC and thrombosis in MPNs? This information can give a further boost of novelty to this review and provide a new clinical marker of thrombotic risk in the MPNs. I think that this information should be reported in this manuscript. Therefore, I think that this article is suitable for publication after minor revision.
Reply: Thank you for your comment. According to you suggestion, we added recent papers published on this, which you can see cited below in the text and in the table. We added in the manuscript the following sentence in lines 176-179: “Venous thrombosis events were recently associated with a high absolute neutrophil count (ANC) and AMC at diagnosis of MPN. in particular, V617F% ≥ 75% or AMC ≥1.5 × 109/L was found to be strongly associated with a higher risk for venous thrombosis in Post-PV MF.”
Reviewer 3 Report
The authors in the review manuscript describe the role of NLR, LMR, PLR, ALC, AMC in myeloid malignancies. I have found some unuccuracies:
Line 67: largely inflammation, what else?
Line 135: explain what it means?
Line 143: sedimentation of what?
Line 145-148: clearer
Line 149-151: comment examinations (41) and (42) that present different results
Author Response
Point-by-point reply to Reviewers’ comments
Reviewer 3
Comments and Suggestions for Authors
The authors in the review manuscript describe the role of NLR, LMR, PLR, ALC, AMC in myeloid malignancies. I have found some inaccuracies:
Line 67: largely inflammation, what else?
Reply: Thank you very much for your comment. We changed the sentence as follows: “Tumor progression is largely dependent on inflammation, which acts as a major driver together with related oxidative stress, angiogenesis, matrix remodeling, as well as on specific genetic mutations [14,15]”
Line 135: explain what it means?
Reply: Thank you very much for your comment. We change the sentence “Clinical features are similar. Myelofibrosis can present either splenomegaly, cytopenias, or overlapping manifestations that lead to a more aggressive disease if erythrocytosis is associated with PV or thrombocytosis with ET” in “Clinical features are similar. Myelofibrosis can present either splenomegaly, cytopenias, or overlapping manifestations that lead to a more aggressive disease. Erythrocytosis occurs most often in PV while thrombocytosis in the ET.”
Line 143: sedimentation of what?
Reply: Thank you very much for your comment. We added in the text “erythrocyte sedimentation rate”
Line 145-148: clearer
Reply: Thank you very much for your comment. We change the sentence “Although, there were increased laboratory data in patients, there were no differences between those with and without thrombotic events.” in “ No significant differences were found in the patient's cohort between those with and without thrombotic events”
Line 149-151: comment examinations (41) and (42) that present different results
Reply: Thank you very much for your comment. We change “An analysis of 70 patients found no association between thrombosis, PLR and NLR in a cohort of 70 patients [41]. In contrast, a larger cohort of 150 patients evaluated NLR and classical thrombotic parameters in ET. It was reported a high NLR ratio in Jak2 positive patients.” In “An analysis of 70 PV and ET patients found no association among thrombosis, PLR and NLR [41]. However, in a larger group of 150 ET patients, NLR was the best predictor of thrombosis events. It was reported, as well, a high NLR ratio in Jak2 positive patients [42].”
Round 2
Reviewer 1 Report
Thanks for providing the point by point answers to the comments, they are justified and accepted.